# Microbiological Etiology in Patients with IE Undergoing Surgery and for Patients with Medical Treatment Only: A Nationwide Study from 2010 to 2020

**DOI:** 10.3390/microorganisms11102403

**Published:** 2023-09-26

**Authors:** Peter Laursen Graversen, Lauge Østergaard, Marianne Voldstedlund, Malthe Faurschou Wandall-Holm, Morten Holdgaard Smerup, Lars Køber, Emil Loldrup Fosbøl

**Affiliations:** 1Department of Cardiology, Copenhagen University Hospital—Rigshospitalet, 2100 Copenhagen, Denmark; laugeoestergaard@gmail.com (L.Ø.); lars.koeber.01@regionh.dk (L.K.); emil.fosboel@regionh.dk (E.L.F.); 2Department of Data Integration and Analysis, Statens Serum Institut, 2300 Copenhagen, Denmark; mav@ssi.dk; 3Danish Multiple Sclerosis Registry, Department of Neurology, University of Copenhagen—Rigshospitalet, 2600 Glostrup, Denmark; malthe.faurschou.wandall-holm@regionh.dk; 4Department of Cardiothoracic Surgery, Copenhagen University Hospital—Rigshospitalet, 2100 Copenhagen, Denmark; morten.holdgaard.smerup@regionh.dk; 5Department of Clinical Medicine, University of Copenhagen, 2200 Copenhagen, Denmark

**Keywords:** microorganism, infective endocarditis, surgery, microbiological etiology, microbiological characteristics

## Abstract

Microbiological etiology has been associated with surgery for infective endocarditis (IE) during admission, especially *Staphylococcus aureus*. We aimed to compare patient characteristics, microbiological characteristics, and outcomes by treatment choice (surgery or not). We identified patients with first-time IE between 2010 and 2020 and examined the microbiological etiology of IE according to treatment choice. To identify factors associated with surgery during initial admission, we used the Aalen–Johansen estimator and an adjusted cause-specific Cox model. One-year mortality stratified by microbiological etiology and treatment choice was assessed using unadjusted Kaplan–Meier estimates and an adjusted Cox proportional hazard model. A total of 6255 patients were included, of which 1276 (20.4%) underwent surgery during admission. Patients who underwent surgery were younger (65 vs. 74 years) and less frequently had cerebrovascular disease, cardiovascular disease, diabetes, and chronic kidney disease. Patients with *Staphylococcus aureus* IE were less likely to undergo surgery during admission (13.6%) compared to all other microbiological etiologies. One-year mortality according to microbiological etiology in patients who underwent surgery was 7.0%, 5.3%, 5.5%, 9.6%, 13.2, and 11.2% compared with 24.2%, 19.1%, 27,6%, 25.2%, 21%, and 16.9% in patients who received medical therapy for *Staphylococcus aureus*, *Streptococcus* spp., *Enterococcus* spp., coagulase-negative Staphylococci, “other microbiological etiologies”, and blood culture-negative infective endocarditis, respectively. Patients with IE who underwent surgery differed in terms of microbiology, more often having Streptococci than those who received medical therapy. Contrary to expectations, *Staphylococcus aureus* was more common among patients who received medical therapy only.

## 1. Introduction

Infective endocarditis (IE) remains a disease associated with high mortality despite improvements in treatment regimens [1,2], and the incidence of IE is increasing in the Western population [3,4,5,6,7]. Long-term antibiotic therapy is the standard treatment of IE and, in some cases, surgery is required. The current European and American guidelines recommend surgery with a class I recommendation in cases of heart failure and uncontrolled infection and IIa/IIb indications for emboli prevention [8,9,10]. Although the guidelines overall agree that small discrepancies exist, and the American guidelines recommend surgery for *Staphylococcus aureus* (*S. aureus*) IE with a class I recommendation [9], whereas the European guidelines recommend surgery if an early favorable response to antibiotics is lacking [8]. The occurrence of surgery during the first admission for IE varies significantly across European cohorts, ranging from 22 to 52% [5,11,12]. Knowledge on the microbiological etiology of IE is pivotal for optimal treatment and risk assessment. The current knowledge on outcomes and treatment choice (surgery during first admission or medical therapy only) in relation to microbiological etiology is derived from selected cohorts with a relatively small number of patients [13,14,15,16]. Thus, unselected contemporary data describing the microbiological etiology stratified by treatment choice are warranted, especially with a focus on *S. aureus*. The aims of this study were:(i)To compare the microbiological characteristics of IE in an unselected nationwide cohort based on the treatment choice.(ii)To identify factors associated with surgery during admission for IE with focus on microbiological etiology.(iii)To assess the rate of surgery and mortality after discharge according to microbiological etiology and treatment choice.

## 2. Materials and Methods

### 2.1. Data Sources

At birth, Danish citizens are provided with a unique personal identifier, enabling crosslinkage in population-based national registries on an individual level [17]. We used the following registries: The Civil Registration System, which contains information on the date of birth, sex, and migration status. The Danish National Patient Registry containing information on hospitalizations since 1977, including the diagnosis codes reported by a physician at discharge. Furthermore, the registry provides information on primary and secondary diagnosis codes, dates of hospital admissions, and discharges. In 1994, the 10th edition of the International Classification of Diagnosis (ICD) was used to code primary and secondary diagnoses, and surgical procedures were added to the registry from 1996. The Danish Prescription Registry contains information on the dates of prescription redemptions, drug types according to the Anatomical Therapeutic Chemical (ATC) classification system, and drug strengths. The Danish Register of Causes of Death contains information on the primary and secondary causes of death registered by a physician and the corresponding date of death. The Danish national registries are all validated, with a high degree of quality and completeness; the registries were described in detail previously [17,18,19,20].

#### The Danish Microbiology Database (MiBa)

Since January 2010, all departments of clinical microbiology transferred copies of every final report to the Danish Microbiology Database (MiBa), including both positive and negative blood cultures. MiBa functions as a resource for the surveillance of infectious diseases in Denmark, to which physicians have access to. In total, 11 departments of clinical microbiology reported to MiBa during the study period. Detailed information regarding blood cultures from MiBa was previously described [21]. All available blood cultures within 30 days prior to the index date to account for diagnostic delay were derived from this registry.

### 2.2. Study Population

All patients admitted for the first episode of IE between 1 January 2010 and 31 December 2020 were identified using the following ICD-8/10 diagnosis codes for infective endocarditis: 421 (ICD-8), DI33.0, DI38.0, and DI39.8. To increase the accuracy of the IE diagnosis, only patients who were hospitalized for >14 days or died within the first 14 days of admission were included, as descried previously, with a positive predictive value of 90% in the Danish National Patient Registry [22,23]. The study population was stratified according to treatment choice: (i) patients who underwent surgery during the initial admission (procedure codes are available in Appendix A) and (ii) patients who received medical therapy only. The microbiological etiology was identified as a positive blood culture collected within 30 days of the index date and until the discharge date of IE. We ranked the possible microbiological etiologies to identify the most likely primary microorganisms causing IE: (1) *S. aureus*, *Streptococcus* spp., HACEK (*Haemophilus* (not including *Haemophilus influenzae*), *Aggregatibacter*, *Cardiobacterium*, *Eikenella*, and *Kingella*), and *Enterococcus* spp.; (2) coagulase-negative Staphylococci (CoNS); (3) “other microbiological etiologies”; and (4) negative blood cultures. Subsequently, patients were categorized according to the following groups of microorganisms: (1) *S. aureus*, (2) *Streptococcus* spp., (3) *Enterococcus* spp., (4) CoNS, (5) “other microbiological etiologies” (including HACEK and fungi) referred to as “other IE”, and (6) blood-culture-negative IE, referred to as “Negative IE”. The specific bacteria assessed in the parent groups of microbiological etiologies are shown in Appendix A. Patients without available blood cultures within 30 days of the index date were excluded (Figure 1).

### 2.3. Outcomes, Follow-Up, and Comorbidities

The primary outcomes were the distribution of microbiological etiologies according to the treatment group, surgical intervention during initial admission, and all-cause mortality after IE discharge. A secondary outcome was surgery within the first year after IE discharge. Assessing association with surgery during initial admission, we followed patients from the date of admission to the date of death, date of surgery, or date of discharge, whichever came first. When assessing outcomes after IE discharge, patients were followed one year until the first occurrence of emigration, death, surgery after discharge, or 31 December 2021. Comorbidities were defined using primary or secondary diagnosis codes from the Danish National Patient Registry within ten years of the index date. All diagnoses from hospitalizations or outpatient visits were included (Appendix A). Comedication six months prior to the index date was defined as a filled prescription for a subspecific drug group from the Danish Prescription Registry. Hypertension was defined as having at least two prescriptions of antihypertensive medication within six months of the index date. Diabetes was defined as either a prescription for glucose-lowering medication within six months of the index date or a diagnosis code for diabetes.

### 2.4. Statistics

Baseline characteristics were compared according to treatment choice during admission (i) and (ii). Categorical variables were reported in frequencies and percentages and continuous variables with medians and 25–75th percentiles. Comparisons between groups were performed using Pearson’s chi-square test for categorical variables and Wilcoxon rank sum test for continuous variables. Bar charts were used to illustrate the proportion of treatment choice stratified by microbiological etiology and to illustrate the proportion of microbiological etiology by treatment choice.

To assess whether the microbiological etiology of IE was associated with surgery during the initial admission, we used the Aalen–Johansen estimator, in which all-cause mortality was treated as a competing risk. The Gray test was used to determine the overall differences between microbiological etiologies. To further explore the association between microbiological etiology and surgery during admission, we performed a cause-specific Cox proportional hazard model. The following variables were included in the final model: age, sex, cardiac implantable electronic devices (CIEDs), prior valve prosthesis, heart failure (HF), ischemic heart disease (IHD), chronic obstructive pulmonary disease (COPD), liver disease, chronic kidney disease (CKD), diabetes, and malignancy. Age was included as a continues variable with a 10-year increase in age for each unit increase. We tested a potential interaction between sex and microbiological etiology and found no statistically significant interaction. The proportional hazard assumption was tested with accumulated residuals and was found valid for all included factors in the cause-specific Cox model. We assessed the cumulative incidence of heart valve surgery after the initial episode of IE within the first year of IE discharge by treatment choice (i) and (ii) using the Aalen–Johansen estimator. All-cause mortality was treated as a competing risk. The Gray test was used to test the statistical differences between the groups.

Survival after discharge stratified by microbiological etiology was estimated using the Kaplan–Meier estimator according to treatment choice (i) and (ii). Patients who died in-hospital were excluded from the survival analyses (N = 1199). The log-rank test was used to estimate the overall difference between the microbiological etiologies. We used the Cox proportional hazard model to assess the association of microbiological etiology and one-year mortality stratified by treatment choice (i) and (ii). We included the same confounders in the final model as in the cause-specific Cox model. The proportional hazards assumption was tested with accumulated residuals and was found valid for all factors included in the final Cox model in both groups (i) and (ii).

In the sensitivity analyses, patients who underwent surgery during admission (cases) were age- and sex-matched 1:1 with a control population of patients who received medical therapy only. A maximum two-year age difference was accepted between the cases and their corresponding controls. We were unable to match six of the cases and excluded these, leaving a matched population of 2540 IE patients (Figure 1). One-year mortality rates were assessed in the matched cohort with the Kaplan–Meier estimator among patients who survived until discharge. We excluded 155 patients who died during admission and four cases that did not have any matching controls (N = 2234). Statistical significance was set at *p* < 0.05. Data management and statistical analyses were performed using SAS Enterprise 7.1 (SAS Institute, Inc., Cary, NC, USA) and R software version 3 (R Foundation for Statistical Computing, Vienna, Austria).

## 3. Results

We identified 6255 patients with IE and available blood cultures between 2010 and 2020. Surgery during first IE admission was performed in 1276 of the patients (20.4%). Of those, 1184 patients underwent solely left-sided valve surgery (92.8%), 31 underwent right-sided surgery (2.4%), and 61 patients underwent combined left- and right-sided surgery (4.8%) (Appendix A). The median time from admission to surgery was 7 days and 75% of the patients underwent surgery within 15 days from admission.

### 3.1. Baseline Characteristics

Baseline characteristics of the study population stratified by treatment choice are shown in Table 1. Patients who underwent surgery during admission were younger (65 vs. 74 years), had a higher proportion of males (76% vs. 65%), and had a longer length of stay compared with patients who received medical therapy only. The proportion of patients with prior prosthetic heart valves was similar in the two groups. Patients who underwent surgery had more aortic and mitral valve disease prior to the index of IE. Otherwise, patients who received medical therapy only had a higher burden of cerebrovascular disease, dementia, cardiovascular disease, diabetes, and chronic kidney disease. Further, patients who received medical therapy only had a higher Charlson comorbidity index score. Similar patterns were observed in the age- and sex-matched cohort (Appendix A). Patient characteristics among those who underwent surgery during admission stratified by the microbiological group of etiology showed that patients with *S. aureus* IE and CoNS IE had a higher proportion of CKD and treatment with dialysis prior to admission. Patients with *Enterococcus* IE were older and had a higher burden of comorbidities, followed by CoNS IE and blood-culture-negative IE (Appendix A).

### 3.2. Microbiological Characteristics and Surgery

Overall, the most common microbiological etiology was *S. aureus* (30.8%), followed by *Streptococcus* spp. (28.0%) and *Enterococcus* spp. (15.7%). Within the etiological groups, patients with *S. aureus* IE had the lowest proportion of surgery during admission (13.6%) compared with the other etiological groups of IE. *Streptococcus* spp. IE had the highest proportion of surgery during admission (25.5%, Figure 2). Across treatment groups, *S. aureus* IE was the most frequent in patients who received medical therapy only (33.4%) in contrast to *Streptococcus* spp. IE in patients who underwent surgery during admission (35.0%). The proportion of negative IE was higher among patients who underwent surgery (16.8% vs. 12.9%). *Enterococcus* spp. IE, CoNS IE, and “Other IE” were evenly distributed between the two groups (Figure 3). In a sensitivity analysis, patients with CIEDs at baseline were excluded, and the same patterns were observed: *S. aureus* IE still had the lowest proportion of surgery during admission (15.5%). The proportion of surgery across all microbiological etiologies increased (Appendix A). Furthermore, the same distribution of microbiological etiologies according to treatment choice was observed (Appendix A). In an age- and sex-matched cohort, we observed the same distribution of microbiological etiologies according to treatment choice, although the difference in the proportion of *S. aureus* IE increased (Appendix A).

### 3.3. Factors Associated with Surgery during Initial Admission for IE

The crude cumulative incidence of initial surgery during admission was lowest for *S. aureus* after 50 days (14.4% (95%CI: 12.8–16.2%)) compared with all other microbiological etiologies. *Streptococcus* spp. had the highest cumulative incidence of surgery after 50 days (27% (CI95%: 24.8–29.4%), Appendix A). In the adjusted cause-specific Cox model, *S. aureus* was significantly associated with a 48% reduced rate of surgery during initial admission compared with *Streptococcus* spp. (HR 0.52 (95% CI: 0.44–0.61)). No statistically significant associations with surgery were found for other microbiological etiologies. Other factors significantly associated with a reduced surgery rate were increasing age, CIEDs, prior valve prosthesis, liver disease, CKD, and malignancy. In contrast, male sex was significantly associated with an increased rate of surgery during admission compared to the female sex (Figure 4).

### 3.4. In-Hospital Mortality Stratified by Microbiological Etiology and Treatment Choice

The proportion of in-hospital mortality among patients who underwent surgery during admission was 2.5% compared to 16.7% in patients receiving medical therapy only. Of all IE patients, *S. aureus* IE had the highest proportion of in-hospital mortality (27.9%) driven by patients who did not undergo surgery (25.4%) compared to the other microbiological etiologies of IE. The proportion of in-hospital mortality in patients who underwent surgery during admission was similar across microbiological etiologies, accounting for approximately 2–3% of all patients with IE. Patients with *Streptococcus* spp. IE had the lowest in-hospital mortality rate (12.0%; Figure 5). For a more detailed overview of the status at discharge (alive or dead), please see Appendix A.

### 3.5. Surgery after Initial Episode of IE

The overall cumulative incidence of heart valve surgery after the initial episode of IE was 3.5% (95% CI: 3.0–4.0%) after 12 months of follow-up from IE discharge. The cumulative incidence in patients who underwent surgery during admission was 2.6% (95% CI: 1.8–3.7%) In contrast, the cumulative incidence was 3.7% (95% CI: 3.2–4.4%) in patients who received medical therapy only (unadjusted *p* = 0.066, Figure 6).

### 3.6. Twelve-Month Mortality Rates after IE Discharge in Patients Surviving the Initial Admission Stratified by Microbiological Etiologies and Treatment Choice

The 12-month mortality rates among patients who underwent surgery and survived the initial admission stratified by microbiological etiology were as follows: *S. aureus* IE, 7%; *Streptococcus* spp. IE, 5.3%; *Enterococcus* spp. IE, 5.5%; CoNS IE, 9.6%; other IE, 13.2%; and negative IE 11.2% (*p* = 0.038, Figure 7A). In comparison, patients who received medical therapy had the following rates: *S. aureus* IE, 24.2%; *Streptococcus* spp. IE, 19.1%; *Enterococcus* spp. IE, 27.6%; CoNS IE, 25.2%; other IE, 21%; and negative IE, 16.9% (*p* < 0.001, Figure 7B). In the sensitivity analyses, including age- and sex-matched cohorts, similar mortality rates were found (Appendix A). In the adjusted analyses, other IE (HR, 2.38 (95%CI: 1.08–5.27)) and negative IE (HR, 2.24 (95%CI: 1.22–4.10)) were associated with increased rates of one-year mortality compared to *Streptococcus* spp. (REF) in patients who underwent surgery during admission. In patients who received medical therapy only, *S. aureus* IE (HR, 1.46 (95%CI: 1.22–1.75)), CoNS IE (HR, 1.62 (95%CI: 1.20–2.15)), and *Enterococcus* spp. IE (HR, 1.36 (95%CI: 1.12–1.67)) were associated with increased rates of one-year mortality.

## 4. Discussion

The main objectives of this study were to compare microbiological characteristics, identify factors associated with surgery during admission, and assess outcomes (rates of surgery and mortality after discharge) according to treatment choice (surgery vs. medical therapy only). The study population consisted of an unselected nationwide cohort of patients with infective endocarditis. This study had three major findings. First, microbiological etiology differed according to treatment choice: Streptococci were most frequent in patients who underwent surgery compared to *S. aureus* in patients who received medical therapy only. Second, in the adjusted analysis, *S. aureus* IE was significantly associated with reduced rates of surgery during admission compared to *Streptococcus* spp. Third, one-year mortality rates differed according to treatment choice and microbiological etiology.

Patients with *S. aureus* and CoNS IE who underwent surgery had a higher proportion of chronic kidney disease and more frequently underwent dialysis prior to IE admission. *S. aureus* and *Enterococcus* spp. Had a longer length of hospital stay. Furthermore, patients with *Enterococcus* spp. And CoNS IE had a higher burden of comorbidity and patients with *Enterococcus* spp. Were older compared to all other microbiological etiologies, in alignment with previous findings [24]. *S. aureus* was the most frequent cause of IE overall, in agreement with previous findings [5,7,24]. Notably, *S. aureus* IE had the lowest proportion of surgery during initial admission compared with all other microbiological etiologies. *S. aureus* IE accounted for one-fifth of cases across the microbiological etiologies in patients who underwent surgery compared to one-third in patients who received medical therapy. *Streptococcus* spp. IE accounted for more than one-third of the cases across microbiological etiologies in patients who underwent surgery during admission. Similarly, Lalani et al. observed that the proportion of *S. aureus* IE was highest among patients who received medical therapy compared with *Streptococcus viridans* IE in patients who underwent surgery [13], consistent with other studies on the same subject [14,16,25,26].

Several factors might have contributed to these findings. First, we included device-related IE, which has been associated with S. aureus IE, thus, partly explaining the lower proportion of surgery among *S. aureus* IE. However, in a sensitivity analysis excluding CIEDs at baseline, *S. aureus* IE still had the lowest proportion of surgery during admission compared to all other microbiological etiologies. Second, *S. aureus* has been associated with increased in-hospital mortality [24,27], and a higher proportion of patients with *S. aureus* IE who did not undergo surgery died during admission in our study (25.4%). The high virulence of *S. aureus* potentially leading to hemodynamic, IE related-complications, or end-organ complications, and the severe comorbidity burden in these patients can result in conservative treatment because of an unacceptably high surgical risk [26,28], despite the indication for surgery according to current guidelines [8,9]. Furthermore, *S. aureus* IE has been strongly associated with healthcare-associated infections, particularly hemodialysis [24,29,30], and previous studies have reported that patients treated with hemodialysis were less likely to undergo surgical treatment [30]. In contrast, early diagnosis in patients with *S. aureus* IE due to a higher proportion of healthcare-associated infections and earlier onset of symptoms leading to an earlier initiation of correct medical therapy, thus, preventing end-organ- and IE-related complications, such as stroke [31], valve destruction, and heart failure, could have resulted in the reduced use of surgery in patients with *S. aureus* IE.

Diagnostic delay has been associated with a higher volume of surgical treatment, especially evident in low virulence *Streptococcus* spp. Due to a higher degree of valve destruction and heart failure [28]. Further, a lack of symptoms often delays the diagnosis of IE, which might explain the higher proportion of negative IE among patients who underwent surgery due to more diffuse symptoms and a lack of fever [32]. Further, negative IE patients could undergo surgery to increase the likelihood of determining the microbiological etiology with a polymerase chain reaction (PCR) of extracted valve tissue or histopathological evaluation, including culture and Gram staining [33]. We found that *S. aureus* was associated with reduced rates of surgery during admission compared to Streptococci, similar to what has been found in previous studies [26,34]; however, we did not have information on important risk factors, such as the indication for surgery, preoperative risk assessment (EuroSCORE II/the Society of Thoracic Surgeons (STS) risk score), and reason for abandoning surgery; thus, we might have overestimated the true association of *S. aureus* IE and the rate of surgery during initial admission compared to other microbiological etiologies. However, Chu et al. found similar odds for surgery during admission, including the STS risk score in quantiles [26].

The one-year mortality after discharge was higher in patients who received medical therapy only than in those who underwent surgery, independent of microbiological etiology, explained by older patients with a higher comorbidity burden among those who received medical therapy only. Further, the one-year mortality patterns stratified by microbiological etiology differed according to treatment choice. The etiology groups “Other” and negative IE cases were associated with the highest one-year mortality among patients who underwent surgery, which might have been due to delayed or incorrect antibiotic treatment because the microbiological etiology was never found. Among patients who received medical therapy only, *Enterococcus* spp. IE, CoNS IE, and *S. aureus* IE were associated with the highest one-year mortality, as reported previously [24]. The high virulence of *S. aureus* IE and the associated increased rate of systemic embolization and morbidity can explain the increased mortality rate for *S. aureus* IE [35,36]. Furthermore, *S. aureus* and CoNS IE have been associated with a higher rate of reinfection [37], most likely due to the higher proportion of patients undergoing hemodialysis treatment, leading to repeated bloodstream infections. The high mortality rate for *Enterococcus* spp. could be partly explained by older age and a higher comorbidity burden at baseline. Furthermore, *Enterococcus* spp. IE has been associated with high rates of recurrent IE episodes, which could be explained through the suboptimal eradication of the infection or deficient eradication of primary or secondary foci (e.g., malignancy) [37].

### Limitations

This study had some limitations. First, all data were derived from hospital coding, and data on clinical characteristics, such as symptoms and duration of symptoms, echocardiography findings, including localization (right- or left-sided IE), and size of vegetation, origin of infection, antibiotic treatment during admission, and laboratory findings other than microbiological etiologies, were unavailable. Further, we did not have any information regarding the percentage of methicillin-resistant *S. aureus* (MRSA) in the treatment groups; however, MRSA only accounts for a small proportion, <2%, of all *S. aureus* bacteremia cases in Denmark [38]. In addition, we lacked information on indications for surgery, such as heart failure, uncontrolled infection (e.g., abscess), or embolization, and surgical risk scores, such as the EuroSCORE II or STS risk score. Furthermore, we lacked information on the reasons for withdrawing from surgery. Microbiological characteristics may have differed in the subgroup analyses depending on surgical indications and reasons for abandoning surgery despite indications. Microbiological etiologies were grouped into six categories, in which *Streptococcus* spp., CoNS, and “other microbiological etiologies” consisted of several microorganisms, which might have been unevenly distributed according to treatment choice (surgery during admission vs. medical therapy). Further, CoNS could have been due to contamination, and these cases might have been misclassified as CoNS IE instead of blood-culture-negative IE. Second, data did not include valve cultures or 16S/18S polymerase chain reaction test results, which could have impacted the blood-culture-negative cases. Misclassification is always a concern in registry-based studies; however, the diagnosis codes of IE were validated previously with a positive predictive value of 90% [22,23]. However, we were not able to distinguish between definite and possible IE cases according to the ESC 2015 modified criteria for the diagnosis of IE [8].

## 5. Conclusions

Data from a nationwide cohort of patients with IE revealed that microbiological etiology differed according to treatment choice: Streptococci were the most frequent microbiological etiologies among patients who underwent surgery, while *S. aureus* was the most frequent in patients who received medical therapy only. Patients’ characteristics among those who underwent surgery showed less comorbidity, younger age, and better outcomes. Future studies are warranted to assess microbiological characteristics in subgroups of patients depending on surgical indications or reasons for withdrawal from surgery.

## Figures and Tables

**Figure 1 microorganisms-11-02403-f001:**
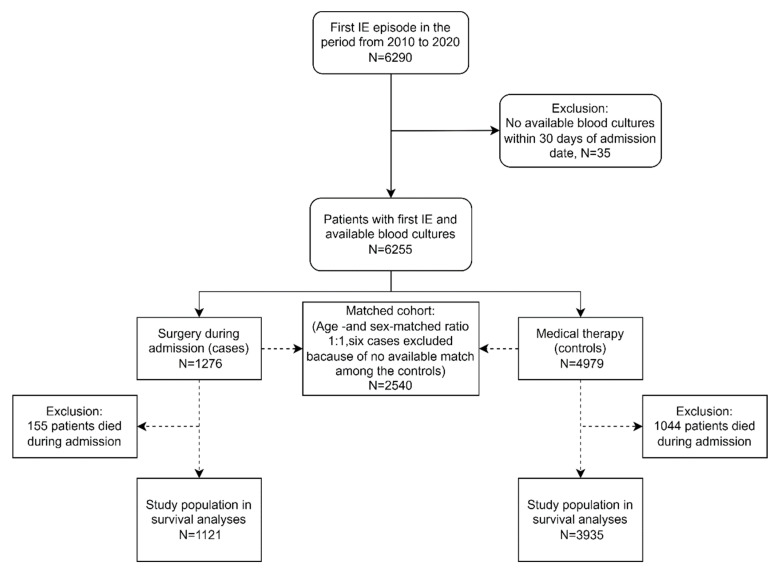
Flowchart of study population. Legend: Matching was performed in a ratio of 1:1 with cases (surgery during admission) and controls (medical therapy). Patients were matched on sex and age (a maximum of two-year difference was allowed in the age-matching). IE—infective endocarditis.

**Figure 2 microorganisms-11-02403-f002:**
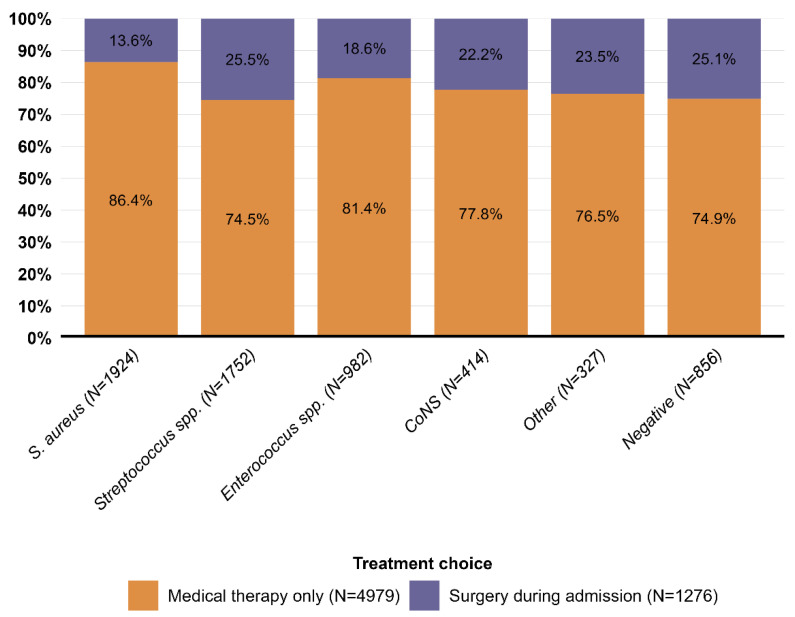
Proportion of treatment choice (surgery during first admission vs. medical therapy only) stratified by microbiological etiology. Legend: The proportion of treatment choice (surgery during admission filled with purple and medical therapy filled with orange) stratified by microbiological etiology. CoNS, coagulase-negative staphylococci; Negative, negative blood culture; Other, “other microbiological etiologies”; *S. aureus*, *Staphylococcus aureus*.

**Figure 3 microorganisms-11-02403-f003:**
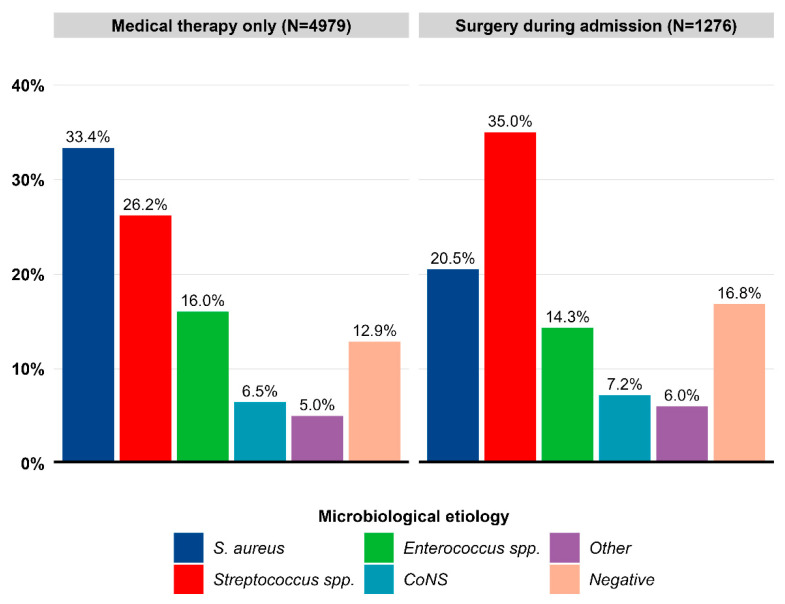
Proportion of microbiological etiologies stratified by treatment choice (surgery during admission vs. medical therapy). Legend: The proportion of microbiological etiology was estimated as the frequency of each microbiological event out of the total number of IE events in the group of patients who underwent surgery (N = 1276) or in the group of patients who received medical therapy (N = 4979). CoNS, coagulase-negative staphylococci; Negative, negative blood culture; Other, “other microbiological etiologies”; *S. aureus*, *Staphylococcus aureus*.

**Figure 4 microorganisms-11-02403-f004:**
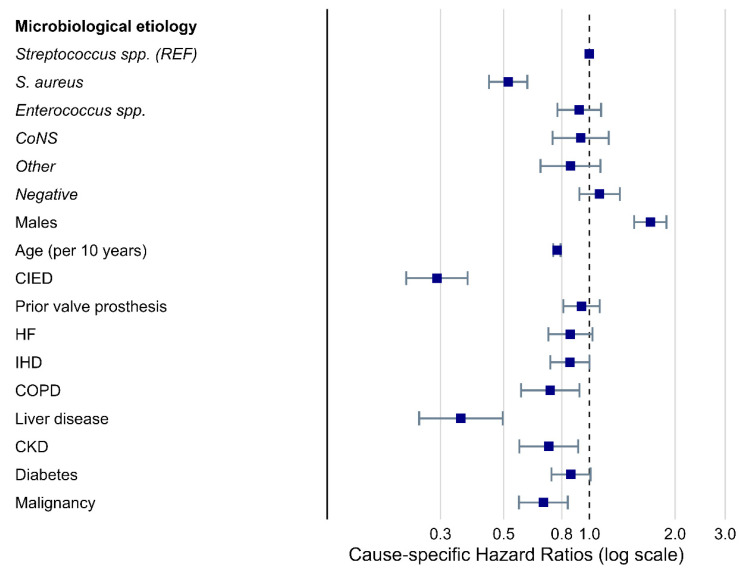
Factors associated with surgery during admission. Legend: Forest plot of cause-specific hazard ratios derived from a multivariate cause-specific Cox model adjusted for microbiological etiology, sex, cardiac implantable electronic devices (CIEDs), prior valve prosthesis, heart failure (HF), ischemic heart disease (IHD), diabetes, malignancy, chronic obstructive pulmonary disease (COPD), and chronic kidney disease (CKD). The outcome was surgery during admission. *Streptococcus* species were used as a reference for microbiological etiology. Age was defined as the cause-specific hazard ratio for surgery during the initial admission for IE for each 10-year increase in age. CoNS, coagulase-negative staphylococci; Negative, negative blood culture; Other, “other microbiological etiologies”; *S. aureus*, *Staphylococcus aureus*.

**Figure 5 microorganisms-11-02403-f005:**
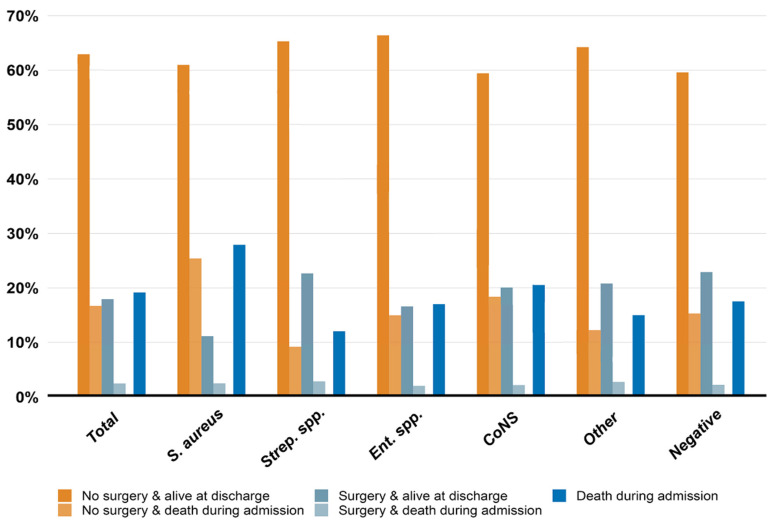
Overview of in-hospital mortality according to microbiological etiology and treatment choice (surgery during admission vs. medical therapy only). Legend: The proportion of patients who died during admission or were alive at discharge stratified by treatment choice and microbiological etiology. The total represents the overall proportion of patients who died during admission or were alive at discharge, including all microbiological etiologies. The blue column represents patients who died during admission, combining patients who received medical therapy and died during admission with those who underwent surgery and died during admission. Please note that the orange-colored and gray-colored columns add up to 100% and represent the entire IE cohort. CoNS, coagulase-negative staphylococci; *Ent*. spp., *Enterococcus* species; Negative, negative blood culture; Other, “other microbiological etiologies”; *S. aureus*, *Staphylococcus aureus*; *Strep*. spp., *Streptococcus* species.

**Figure 6 microorganisms-11-02403-f006:**
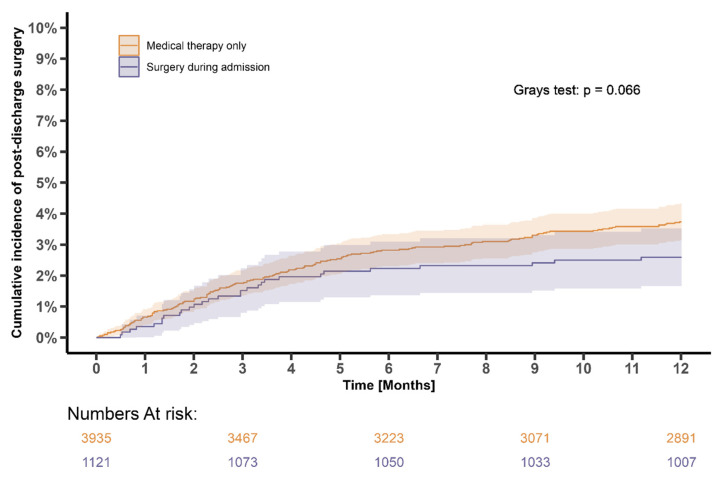
Twelve-month cumulative incidence of surgery after the initial IE stratified by treatment choice (surgery during admission vs. medical therapy). Legend: The Aalen–Johansen estimator was used to depict the 12-month cumulative incidence of heart valve surgery after initial IE stratified by treatment choice, with surgery during admission marked as purple and medical therapy marked as orange. All-cause mortality was treated as a competing risk.

**Figure 7 microorganisms-11-02403-f007:**
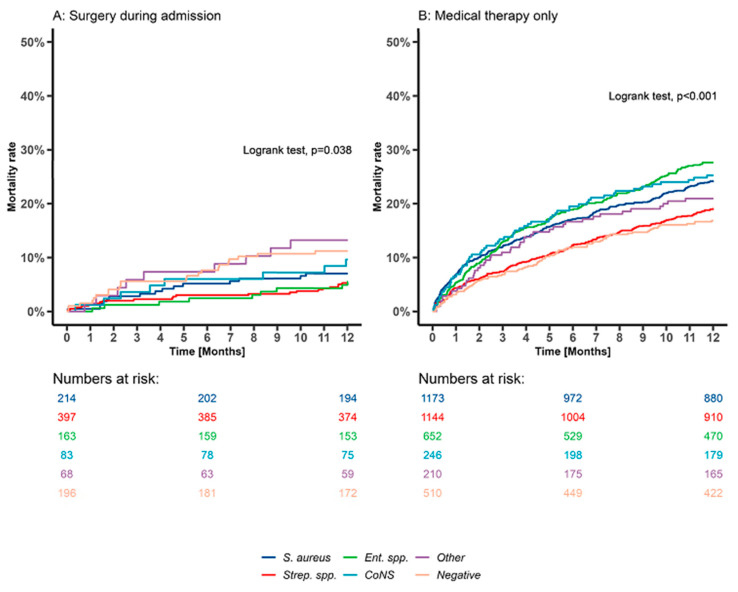
Twelve-month mortality rates after IE discharge by microbiological etiology and treatment choice (surgery during admission vs. medical therapy only) among patients surviving initial admission. Legend: Kaplan–Meier curves stratified by microbiological etiology; (**A**) patients who underwent surgery during admission; (**B**) patients who received medical therapy only. Time since discharge in months for a total of one year follow-up from discharge. *p*-values represent the log-rank test for statistical differences between the survival curves. CoNS, coagulase-negative staphylococci; *Ent*. spp., *Enterococcus* species; Negative, negative blood culture; Other, “other microbiological etiologies”; *S. aureus*, *Staphylococcus aureus*; *Strep*. spp., *Streptococcus* species.

**Table 1 microorganisms-11-02403-t001:** Baseline characteristics stratified by treatment choice (medical therapy only vs. surgery during admission).

Variables	Medical Therapy Only (N = 4979 ^a^)	Surgery during Admission (N = 1276)	*p*-Value ^b^
Males	3238 (65.0)	973 (76.3)	<0.001
Age	74 [65–81]	65 [54–72]	<0.001
Length of hospital stay (days)	35 [24–47]	46 [35–58]	<0.001
Length of hospital stay among patients alive at discharge (days)	40 [30–48]	46 [39–59]	<0.001
Time from admission to surgery (days)	-	7 [3–15]	-
Microbiological etiology ^c^			<0.001
*Staphylococcus aureus*	1662 (33.4)	262 (20.5)	
*Streptococcus*	1305 (26.2)	447 (35.0)	
*Enterococcus*	799 (16.0)	183 (14.3)	
CoNS	322 (6.5)	92 (7.2)	
Other	250 (5.0)	77 (6.0)	
Negative	641 (12.9)	215 (16.8)	
Prior prosthesis	1051 (21.1)	248 (19.4)	0.189
Cardiac implantable electrical devices (CIEDs)	1020 (20.5)	70 (5.5)	<0.001
Aortic valve disease	1224 (24.6)	362 (28.4)	0.006
Mitral valve disease	327 (6.6)	124 (9.7)	<0.001
Atrial fibrillation (AF)	1300 (26.1)	189 (14.8)	<0.001
Heart failure (HF)	1062 (21.3)	156 (12.2)	<0.001
Ischemic heart disease (IHD)	1428 (28.7)	226 (17.7)	<0.001
Cerebrovascular disease (CVD)	760 (15.3)	149 (11.7)	0.001
Dementia	123 (2.5)	≤3 ^d^	<0.001
Hypertension	2813 (56.5)	512 (40.1)	<0.001
Chronic kidney disease (CKD)	510 (10.2)	78 (6,1)	<0.001
Dialysis	293 (5.9)	53 (4.2)	0.004
Diabetes	1249 (24.9)	202 (15.8)	<0.001
Liver disease	290 (5.8)	35 (2.7)	<0.001
Chronic obstructive pulmonary disease (COPD)	581 (11.7)	76 (6.0)	<0.001
Charlson comorbidity index			<0.001
0	1642 (33.0)	709 (55.6)	
1–2	1847 (37.1)	413 (32.4)	
>2	1490 (29.9)	154 (12.1)	
Anticoagulants	1641 (33.0)	260 (20.4)	<0.001
Beta blockers	2072 (41.6)	336 (26.3)	<0.001
Lipid-lowering medication	2116 (42.5)	409 (32.1)	<0.001
RAS inhibitors	2180 (43.8)	464 (36.4)	<0.001

^a^ Median [25–75th percentile] or frequency (%); ^b^ Pearson’s Chi-squared test; Wilcoxon rank sum test; ^c^ all available blood cultures from 30 days before IE admission until date of discharge were collected and ranked as mentioned in the Materials and Methods Section; ^d^ anonymized due to regulation from Statistics Denmark.

## Data Availability

The data presented in this study are available on request from the corresponding author. The data are not publicly available due to restriction from Statistics Denmark.

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
