# Peer review of "Microbiological Etiology in Patients with IE Undergoing Surgery and for Patients with Medical Treatment Only: A Nationwide Study from 2010 to 2020"

_microorganisms, 2023, doi:10.3390/microorganisms11102403_

Round 1
Reviewer 1 Report
The study has several important methodological flaws:
1. there was no information on indications for surgery
2. there was no information on surgical risk assessment according to EuroSCORE II or the Society of Thoracic Surgeons (STS) risk scale
3. there was no information about the reasons for abandoning the operation
4. it is possible that high virulence of S. aureus and significantly more numerous and severe comorbidities in patients with S. aureus infective endocarditis (IE) cause more severe complications that require surgery, but are also associated with its unfavorable outcome. Therefore, it is possible that conservative treatment is the result of an unacceptably high operative risk, and not the absence of indications for surgical treatment.
The mentioned flaws significantly compromise the conclusions/messages of the study:
1) S. aureus IE had the lowest proportion of surgical interventions during initial admission compared to patients with all other groups of microbiological etiology.
2) in adjusted analysis, S. aureus IE was significantly associated with reduced odds of surgery during admission
3) one-year mortality rates differed depending on the choice of treatment and microbiological etiology; among patients who received only medicinal therapy, S. aureus is one of the most common causative agents.
Finally, the main message of the study that data from an unselected national cohort of IE patients reveal that those requiring surgery differ in terms of microbiological etiology, with more cases having streptococci than patients receiving medical therapy alone, is not scientifically sound.
..
Reviewer 2 Report
In this manuscript entitled “Microbiological etiology in patients with IE undergoing surgery and for patients with medical treatment only: A nation-wide study from 2010 to 2020” 6255 patients with first episodes of endocarditis with available blood culture data that occurred between 2010 and 2020 were analyzed. The data presented are interesting and unque. They are extensive and multiple subsets of patients are examined. As a result, the results are somewhat difficult to follow as described. It is unclear whether this relate to structure/organization (I would consider describing “Results 3.5. In-hospital mortality stratified by microbiological etiology “ before addressing the data regarding events after discharge) or a language issue Danish to English.
According to the Introduction the following are stated the aims:
i) To compare the microbiological characteristics of IE in an unselected nation-wide cohort based on the treatment choice.
ii) To identify factors associated with surgery during admission for IE with focus on microbiological etiology.
iii) To assess the rate of surgery and mortality after discharge according to micro-biological etiology and treatment choice
The subsequently stated (section 2.3) “primary outcome was all-cause mortality after IE discharge. A secondary outcome was surgery within the first year after IE discharge. These seems to address only “iii” above.
In “2.4 Statistics” paragraph 2; the text states “To assess whether the microbiological etiology of IE was associated with surgery during admission, A multivariable logistic regression analysis was conducted with adjustment for potential confounders. The analysis was restricted to patients who survived until dis-charge (N=5,056).” This stated goal seems to be addressed by in Table 1 and Figures 2 and 3 which includes all cases and in the “sensitivity analysis” where match cohorts drawn from all patients were analyzed . To address to the stated goal and exclude the 1199 patients not surviving initial admission does not seem correct.
Re above, if I understand correctly what is presented in “3.3. Association with surgery during the first admission” these data refer to the so called “sensitivity analysis “and should be so designated rather than referring vaguely to the “adjusted analysis”.
In “Statistics 2.4” paragraph 3: Survival stratified by microbiological etiology was estimated using the 1-Kaplan-Meier estimator according to treatment choice(i,ii). Patients who died in-hospital were excluded from survival analyses (N=1,189). The “survival” might more clearly be referred to as “ survival after discharge” Also,the number exclude would from Figure 1 seem to be 1199 (155 +1044).
In “Statistics 2.4” paragraph 4: how the populations to be compared in the sensitivity analyses are developed is a bit unclear are requires going back to Fig 1. Perhaps, this could be clarified. What is this meant to convey “(N=2,236, 155 died in-hospital, and four patients had no match).”
In the adjusted analyses -multivariable logistic regression and Cox proportional hazards model - were all of the variables listed included in the final analyses /model or just those with a specific minimum p value?
In Results 3.5. In-hospital mortality stratified by microbiological etiology. The presumed Figure 6 is mislabeled Figure 5. Even with this figure these data are hard to follow. For example, in
“ The proportion of in-hospital mortality among patients who underwent surgery during admission was 2.5% compared to 16.7% in patients receiving medical therapy only.” I would have thought mortality in surgically treated patients was 155/1276 (12.1%) or if you mean the proportion of all deaths that were among surgically treated it would seem be 155/1199 ( 12.9%). The numbers (vs the percentages) for these data are not in the text and the convention in the figure (Please note that the orange-colored columns and gray-colored columns add up to 100%.) are hard to follow it is hard for the reader to assess the related comments in the text. Perhaps this could addressed by presenting the data in a table rather than a figure or by referencing a table in the supplement.
Results 3.6. Mortality rates after IE discharge stratified by microbiological etiologies. In addition to the section heading, it would be helpful to the reader to specify to the reader that in the text and relevant Figure 7A and 7B, these are mortality rates at 12 months among patients surviving the initial admission , e.g. In patients who underwent surgery during admission, the 12 months mortality rates among patients surviving initial admission…Comments regarding 7B should be similarly clarified.
Limitations: There is a typo: “Microbiological etiologies were grouped into sex categories …”. I think this should be six rather than sex
Conclusions: “Data from an unselected nationwide cohort of patients with IE revealed that those who require surgery differ in terms of microbiological etiology, with more cases having streptococci than patients who received medical therapy only.” This statement is neither clear or totally correct. Given the limitation noted- absence of data regarding indications for surgery (and surgical risk assessment) you are not reporting on patients who “require” surgery per current indications but rather those who undergo surgery. Undoubtedly, some patients wherein surgery was indicated refused surgery or were judged at too high risk and were treated with medical therapy alone. Also, your point about distribution of patients with IE caused by streptococci is not very clear.
There are some very unclear sentences. In other places the meaning of the sentence is not ideally clear. Although the meaning can be discerned with additional effort, the readability of the manuscript can be significantly improved by revision. Examples are cited in comments to the authors
Round 2
Reviewer 1 Report
...
....
Author Response
No comments from Reviewer 1
Reviewer 2 Report
The revised manuscript is notably better structured and thus more logical. It is now very readable. It is unclear to me whether an "immortality bias"is operative, as indirectly suggested by your Discussion comments, in the differences in surgical intervention among patients with endocarditis caused by S. aureus vs. streptococci or whether population based data such as used here are sufficiently granular to fully assess this consideration. Hopefully future research will help define those patients where surgical indication are clearly evident and who would truly benefit from surgical treatment (survive) in spite of Euroscores suggesting high surgical mortality risk. Depriving these high risk patients of surgery is clearly associated with increased mortality.
There are a few typographic errors in the text: line 84 - "emboly" should be emboli, line 193 "continues" should be continuous, line 280 "surery " appears as if it should be surgery. The sentence beginning on line 319 is unclear. It appears to be a run-on sentence that likely is meant to be two sentences.
In the interest of space, figure 3 could be deleted since those data are clearly evident in Table 1.
With the minor exceptions noted above the quality of English Language is fine.
